# Power of Augmented Replicas in Out-Of-Distribution Detection

## Abstract

Data augmentation is widely used in machine learning to enhance training datasets by introducing minor variations to the original data, traditionally aiming to prevent overfitting and improve model performance. This paper explores a novel application of data augmentation during the inference stage to enhance out-of-distribution (OOD) detection. The proposed method involves replicating the inference image multiple times, applying various transformation techniques to each replica, and then evaluating the detectors using these augmented images. The effectiveness of this approach is assessed across different detectors, models, and datasets, demonstrating its potential to improve OOD detection capabilities.

## 1 Introduction

In the current era of Artificial Intelligence (AI), Machine Learning (ML) models play a crucial role in numerous applications ranging from computer vision to natural language processing. However, despite their success, these models often face inputs for which they have not been trained (i.e., uncertainty). Out-of-distribution (OOD) data, which are different from training data in terms of distribution, can lead to erroneous and unreliable predictions, which in turn can have detrimental consequences in critical applications (e.g., functional safety-related systems). Therefore, OOD data detection has become an important area of research in the trustworthy ML field.

OOD detectors are tools that can identify if a given input falls outside the distribution of the training data. They are crucial in ensuring that the model performs well (in the sense of reducing false-positives) not just on the data that contains images of the training classes, but also on new, unseen and unknown. If we perform a thorough review of the state of the art, we observe that different detectors are proposed (see Section 4). The standard approach for OOD detection involves generating a scoring function from the trained network such that the In-Distribution (ID) samples show significantly different scores than OOD.

Data augmentation is a technique commonly used in ML to increase the amount of training data while diversifying the *"view"* of an input to which a model is exposed. It consists of creating new data samples conceptually class agnostic, from existing ones by applying transformation techniques (e.g., rotation, scaling, horizontal/vertical flipping). It is commonly used during model training with the aim of preventing overfitting, increasing data diversity and quantity, improving model performance and compensating for missing data.

In this paper, we propose employing augmentation during inference and combine it with OOD detectors to increase the models' degree of trustworthiness. Unlike the traditional use of data augmentation, which is to use it only in training, our approach is based on replicating the inference image a number of times and applying transformation techniques to each of the replicas. In this way, we can evaluate each replica on the detector(s) and observe if the results are consistent with small perturbations of each input. We evaluate this approach on different detectors, models and datasets, in order to know if this can improve the detection of OODs. For this purpose, we have taken as a base work the articles of Sun et al. (2022) and Park et al. (2023).

The remainder of this publication is organized as follows. Section 2 describes the proposed approach. Section 3 performs the evaluation of the contribution in two different scenarios. Section 4 performs a review of the available literature. Finally, Section 5 draws the obtained conclusions.

## 2 APPROACH DESCRIPTION

In this study, we propose a novel approach to improve OOD detection using multiple transformed instances of a single image. Our approach relies on the principle of internal consistency. Similar to how psychological tests use varied phrasing of similar questions to enhance accuracy, completeness, and reliability, our proposed method evaluates the consistency of detector results when subjected to sets of transformed inputs. In other words, it reduces response bias and ensures that responses are not influenced by the interpretation of a single question (or input image, in our case).

Thus, just as psychologists ask the same question, rephrased differently, we evaluate the model with the same original image but replicated and transformed in a different way. Hence, allowing to judge consistency of response. Input transformations cannot be either too small, as this would only result in an increase of resource usage without any benefit, or too much, as this would only create a significant reduction of the classification capabilities of the model. In this paper, we propose using common augmentation methods for the input transformations. Note that what actually constitutes an optimal augmentation level is still an open question.

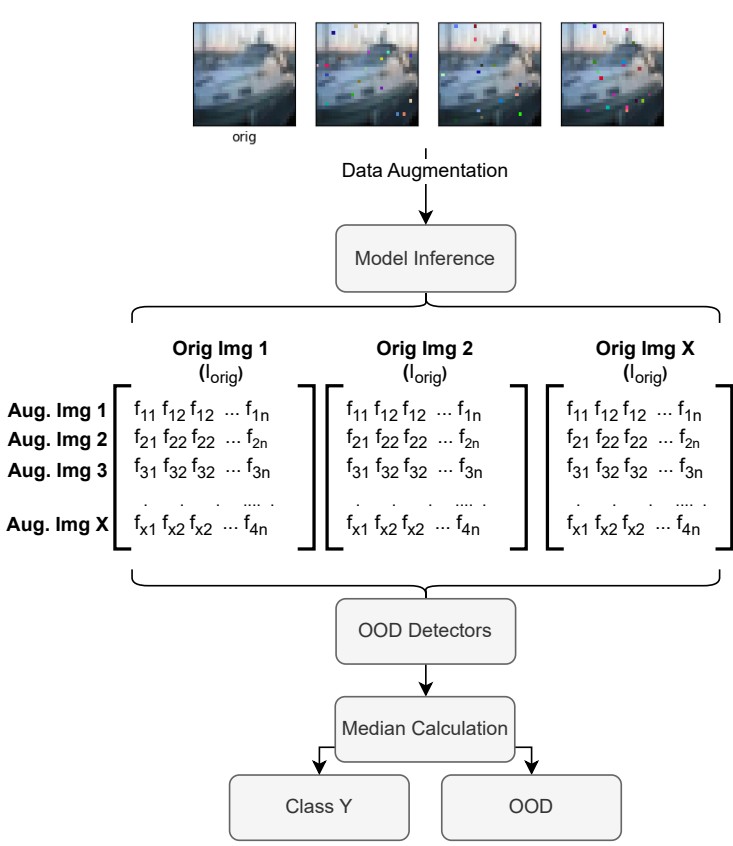

Figure 1: Process of the proposed approach.

The process of the proposed approach (see Figure 1) is as follows:

1. Image Replication: We start with an original image, denoted as $I_{orig}$, and create $X$ copies of it. This results in a set of images $I_1, I_2, ..., I_X$, where each of them is identical to the original.

2. Random Transformations: Each image $I_i$ is then subjected to a random augmentation, obtaining $X$ augmented images $I'_i = T_i(I_i)$.

3. Model Inference: We then perform inference using a pre-trained model on each transformed image $I_i'$, resulting in a set of feature vectors $F_1, F_2, ..., F_X$ .

4. OOD Detection: Each feature vector $F_i$ is provided to an OOD detector $D$, which outputs a score.

$$S_i = D(F_i)$$

The score $S_i$ indicates the degree to which the model believes the image $I_i'$ to be out-of-distribution.

5. Median Calculation: Finally, we calculate the median of all scores $S_i$ to obtain a final OOD verdict for the original image $I_{orig}$. This is given by

$$S_{median} = median(\{S_1, S_2, ..., S_X\})$$

.

We calculate the median using the results obtained by the detectors. That is, we calculate the median with the results of each augmented image. This results in a median score for each original image.

This approach leverages the power of ensemble methods and robust statistics (the median) to provide a more reliable and robust OOD detection mechanism. Future work could explore different types of transformations and models, as well as other statistical measures beyond the median.

From a computational point of view, our approach leads to an increase in processing time and a higher use of hardware resources as it needs to perform inference on a higher number of images. Nevertheless, as there are use cases (e.g., medical image processing) where an increase of this magnitude is not a showstopper, the overhead seems tolerable. Note though that in this research we do not perform any timing or performance examination.

## 3  EVALUATION

In this section, we evaluate the performance of the OOD detectors when combined with our approach. Specifically, we assess the detectors by applying them to augmented replicas. For this purpose, we calculate the median value of the detector results for the replicated images. For example, if we have 8 replicas, we observe the score provided by the detector for these 8 cases and compute the median result. By calculating the median of the replicates, we can find the midpoint of the same varied question, i.e., the same image is transformed differently.

Note that the scripts used to perform the evaluation are published in Unknown (2024) and are based on code from Sun et al. (2022) and Park et al. (2023).

**OOD Detectors**. The analysis is performed using various differential detectors identified during the literature review: Energy by Liu et al. (2020), Nearest Neighbor Guidance (NNGuide) by Park et al. (2023), Maximum Softmax Probability (MSP) by Hendrycks & Gimpel (2016), MaxLogit by Wei et al. (2022), Self-Supervised Outlier Detection (SSD) by Sehwag et al. (2021), Mahalanobis by Lee et al. (2018), and K-th Nearest Neighbor (KNN) by Sun et al. (2022). For further information on these detectors, see Section 4.

**Metrics to Evaluate the Performance of the Approach**. To validate our approach, we examine two metrics: (i) the false positive rate for OOD instances when the true positive rate for ID instances is set at 95% (i.e., False Positive Rate at 95% True Positive Rate (FPR95)), and (ii) the Area Under the Receiver Operating Characteristic Curve (AUROC). These metrics are widely used in OOD detection to evaluate the performance of models for distinguishing between ID and OOD (as in Sun et al. (2022); Park et al. (2023); Hendrycks & Gimpel (2016); Wei et al. (2022)). In OOD detection, lower values of FPR95 indicate superior performance, as they represent fewer negative samples being incorrectly classified as positive when the true positive rate is set to 95%. Conversely, higher values of AUROC indicate higher performance, as they reflect a greater ability of the model to distinguish between in-distribution and out-of-distribution samples. Consequently, for our approach to be considered valid, FPR95 must decrease and AUROC should increase as the number of augmented replicas increases.

**Diverse Evaluations**. To ensure the robustness of our contribution, we perform the same analysis in different scenarios. This involves using various training data (i.e., ID datasets with different sizes),

different models (e.g., MobileNetV2, ResNet), different data augmentation techniques (e.g., Pixel Flip, AugMix), and different datasets as OOD. The analysis is conducted in two scenarios. In the first scenario, we use a ResNet18 model trained with the CIFAR-10 dataset. In the second scenario, we employ a MobileNetV2 model trained with the MedMNIST PathMNIST dataset by Yang et al. (2023) .

## 3.1 ID: CIFAR-10 AND RESNET18

**Model architecture**. For the first experiment, the model employed is a ResNet18, specifically the one used by Sun et al. (2022). The model is trained with CIFAR-10 dataset by Krizhevsky & Hinton (2009), that are sized $3 \times 32 \times 32$. The only difference between the model of Sun et al. (2022) and this one, is that this model has been trained with some additional transformation technique (i.e., random pixel flip). The model has been trained through 100 epochs and results in an accuracy of 94.24%.

**OOD datasets**. In addition, it is examined with different OOD datasets, both near OOD and far OOD. Street View House Numbers (SVHN) (Netzer et al., 2011), Describable Textures Dataset (DTD) (Cimpoi et al., 2014), Places365 (Zhou et al., 2017), iSUN (Xu et al., 2015), Large-scale Scene UNderstanding (LSUN) (Yu et al., 2015) and CIFAR-100 (Krizhevsky & Hinton, 2009). The latter is considered near OOD as images are similar or even collisions (e.g., trucks in Cifar-10 and vehicles in Cifar-100, which contains some trucks).

**Augmentation techniques**. The model is trained with traditional augmentation techniques such as Random Crop and Random Horizontal Flip. Besides, in order to increase diversity, we add some noise to the images by changing the color of randomly selected pixels (see Figure 1). Note that the transformations applied during the training process are also utilized during inference. In other words, the same augmentation techniques are applied during both training and inference.

**Obtained results employing augmented replicas**. Firstly, in this preliminar analysis we inspect if the use of augmented replicas improves detection. In other words, we examine if infering higher number of transformed images reduces FPR95 and increases AUROC values. Table 1 depicts the FPR95 obtained with different replica numbers. All scenarios, i.e., datasets and detectors, show an enhancement in increasing the number of augmented replicas. In other words, calculating the median of the scores that belong to the same image transformed differently through several replicas improves the detection of OODs. Some detectors show a more significant improvement than others, but in all cases, a decrease of the FPR95 metric is observed. Furthermore, in the case of AUROC analysis, equivalent results are obtained. Table 2 shows that increasing the number of replicas improves the AUROC result in all the cases, with no exception. In conclusion, the improved AUROC and FPR95 results indicate that the use of our approach improves the detection of OOD.

Table 1: FPR95 values for different detectors, datasets and number of replicas (i.e., 1, 8 and 32). Results obtained with ResNet18 model trained with CIFAR-10 dataset.

| detectors | SVHN | | | iSUN | | | CIFAR-100 | | | LSUN | | | DTD | | | Places365 | | |
|---|---|---|---|---|---|---|---|---|---|---|---|---|---|---|---|---|---|---|
| | 1 | 8 | 32 | 1 | 8 | 32 | 1 | 8 | 32 | 1 | 8 | 32 | 1 | 8 | 32 | 1 | 8 | 32 |
| energy | 15.07 | 13.08 | 12.71 | 29.71 | 26.57 | 26.14 | 51.05 | 49.51 | 48.72 | 17.22 | 15.36 | 14.79 | 25.62 | 15.78 | 15.37 | 21.29 | 19.91 | 20.00 |
| nnguide | 16.14 | 14.35 | 13.97 | 39.88 | 36.80 | 35.98 | 60.76 | 60.17 | 59.09 | 23.74 | 22.27 | 21.46 | 18.14 | 11.81 | 11.13 | 27.85 | 26.69 | 26.19 |
| msp | 37.30 | 29.68 | 27.44 | 52.18 | 45.61 | 43.45 | 63.08 | 57.75 | 56.38 | 39.98 | 33.80 | 31.74 | 70.48 | 54.75 | 51.12 | 44.25 | 38.16 | 36.80 |
| maxlogit | 16.23 | 14.08 | 13.18 | 31.60 | 28.50 | 26.81 | 51.43 | 49.79 | 48.37 | 18.60 | 16.64 | 15.59 | 29.34 | 17.22 | 16.81 | 22.75 | 21.29 | 20.73 |
| ssd | 15.74 | 14.68 | 14.06 | 94.25 | 94.75 | 94.94 | 78.50 | 78.65 | 77.97 | 49.17 | 48.65 | 47.82 | 23.58 | 7.98 | 7.93 | 51.98 | 50.67 | 49.80 |
| mahalanobis | 13.87 | 12.40 | 11.99 | 90.03 | 90.33 | 90.17 | 74.47 | 73.70 | 72.95 | 44.40 | 42.76 | 41.94 | 21.67 | 7.36 | 7.11 | 45.67 | 43.51 | 42.70 |
| knn | 24.73 | 22.56 | 21.37 | 34.40 | 31.32 | 30.59 | 52.23 | 50.40 | 48.81 | 28.71 | 25.86 | 24.91 | 54.38 | 44.61 | 43.58 | 31.62 | 29.88 | 28.96 |

Table 2: AUROC values for different detectors, datasets and number of replicas (i.e., 1, 8 and 32). Results obtained with ResNet18 model trained with CIFAR-10 dataset.

| detectors | SVHN | | | iSUN | | | CIFAR-100 | | | LSUN | | | DTD | | | Places365 | | |
|---|---|---|---|---|---|---|---|---|---|---|---|---|---|---|---|---|---|---|
| | 1 | 8 | 32 | 1 | 8 | 32 | 1 | 8 | 32 | 1 | 8 | 32 | 1 | 8 | 32 | 1 | 8 | 32 |
| energy | 97.17 | 97.54 | 97.63 | 94.36 | 94.97 | 95.12 | 88.03 | 88.88 | 89.10 | 96.75 | 97.15 | 97.27 | 94.89 | 96.52 | 96.66 | 96.10 | 96.37 | 96.43 |
| nnguide | 96.67 | 97.10 | 97.19 | 92.19 | 92.96 | 93.15 | 83.94 | 84.82 | 85.10 | 95.39 | 95.85 | 96.00 | 95.77 | 96.99 | 97.10 | 94.62 | 94.95 | 95.01 |
| msp | 94.85 | 95.69 | 95.94 | 91.99 | 92.94 | 93.17 | 88.11 | 89.08 | 89.35 | 94.36 | 95.16 | 95.41 | 87.22 | 90.20 | 90.77 | 93.75 | 94.43 | 94.58 |
| maxlogit | 97.02 | 97.43 | 97.54 | 94.19 | 94.86 | 95.01 | 88.01 | 88.92 | 89.14 | 96.59 | 97.03 | 97.16 | 94.42 | 96.25 | 96.41 | 95.95 | 96.26 | 96.32 |
| ssd | 96.82 | 97.19 | 97.30 | 67.77 | 68.88 | 69.05 | 75.66 | 76.76 | 77.18 | 88.19 | 88.94 | 89.21 | 95.81 | 97.73 | 97.78 | 88.73 | 89.46 | 89.58 |
| mahalanobis | 97.34 | 97.68 | 97.78 | 76.72 | 78.06 | 78.33 | 79.95 | 81.04 | 81.50 | 91.28 | 91.98 | 92.24 | 95.98 | 97.80 | 97.85 | 91.53 | 92.19 | 92.31 |
| knn | 95.94 | 96.43 | 96.56 | 94.06 | 94.71 | 94.87 | 89.73 | 90.47 | 90.67 | 95.45 | 95.96 | 96.10 | 82.16 | 84.69 | 85.00 | 94.93 | 95.32 | 95.39 |

Figure 2 can be considered a summary of Tables 1 and 2. Figure shows the mean values of all detectors. In other words, we calculate the mean value of detectors with one replica, 8 replicas and 32 in order to examine the tendency of detectors. The graphs show the FPR95 decreasing while AUROC values increase in all datasets. In addition, it is possible to observe that the improvement is not linear. In both metrics, it reaches an asymptote. Hence, although our proposal improves detection, it is also a limited enhancement.

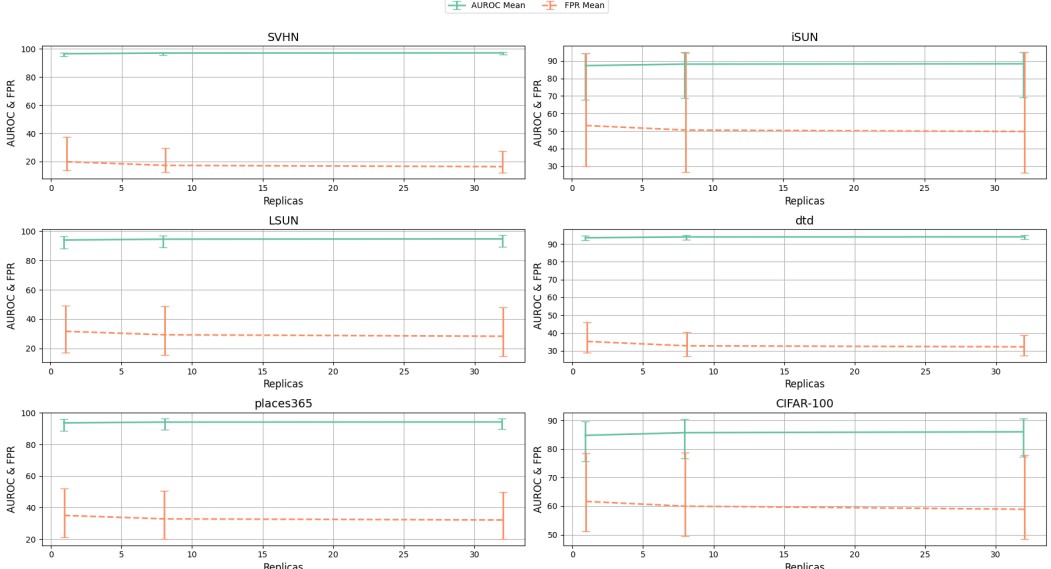

Figure 2: Mean FPR95 and mean AUROC of all detectors through different OOD datasets. Results obtained with ResNet18 model trained for CIFAR-10 dataset.

**Non Augmented vs Augmented inference**. The results presented above compare traditional approaches and the use of large number of augmented replicates. However, this does not indicate that the detection of OOD is necessarily superior to using the detectors as in the current literature. That is, without replication and transformation. Thereby, we compare the results with one augmented image vs non augmented. In other words, we examine the difference between i) traditional OOD analysis without data augmentation during inference and ii) using augmented images during inference.

Tables 3 and 4 provide the obtained results. The comparison is done with 32 replicas and the tables also provides the Improvement column (i.e., Impr.). If there is an improvement the result appears with +. On the contrary, if there is no improvement it appears with − and the value is boxed. Tables 3 and 4 show that there are only a few cases where there is no improvement. Besides, in some of these cases the difference is very small.

Table 3: FPR95 values for different approaches (i.e., no augmentation vs 32). Results obtained with ResNet18 model trained with CIFAR-10 dataset.

| detectors | SVHN no | SVHN 32 | SVHN Impr. | iSUN no | iSUN 32 | iSUN Impr. | CIFAR-100 no | CIFAR-100 32 | CIFAR-100 Impr. | LSUN no | LSUN 32 | LSUN Impr. | DTD no | DTD 32 | DTD Impr. | Places365 no | Places365 32 | Places365 Impr. |
|---|---|---|---|---|---|---|---|---|---|---|---|---|---|---|---|---|---|---|
| energy | 14.25 | 12.71 | +1.54 | 28.71 | 26.14 | +2.57 | 50.33 | 48.72 | +1.61 | 16.43 | 14.79 | +1.64 | 12.36 | 15.37 | −3.01 | 20.81 | 20.00 | +0.81 |
| nnguide | 15.76 | 13.97 | +1.79 | 39.92 | 35.98 | +3.94 | 60.12 | 59.09 | +1.03 | 23.11 | 21.46 | +1.65 | 8.88 | 11.13 | −2.25 | 27.51 | 26.19 | +1.32 |
| msp | 34.10 | 27.44 | +6.66 | 49.94 | 43.45 | +6.49 | 60.60 | 56.38 | +4.22 | 37.73 | 31.74 | +5.99 | 57.82 | 51.12 | +6.70 | 42.31 | 36.80 | +5.51 |
| maxlogit | 15.58 | 13.18 | +2.40 | 30.87 | 26.81 | +4.06 | 50.96 | 48.37 | +2.59 | 17.94 | 15.59 | +2.35 | 14.41 | 16.81 | −2.40 | 22.47 | 20.73 | +1.74 |
| ssd | 15.93 | 14.06 | +1.87 | 94.40 | 94.94 | −0.54 | 79.28 | 77.97 | +1.31 | 49.27 | 47.82 | +1.45 | 45.32 | 7.93 | +37.39 | 53.26 | 49.80 | +3.46 |
| mahalanobis | 14.01 | 11.99 | +2.02 | 90.12 | 90.17 | −0.05 | 74.72 | 72.95 | +1.77 | 44.26 | 41.94 | +2.32 | 44.96 | 7.11 | +37.85 | 46.33 | 42.70 | +3.63 |
| knn | 24.08 | 21.37 | +2.71 | 33.96 | 30.59 | +3.37 | 51.48 | 48.81 | +2.67 | 28.61 | 24.91 | +3.70 | 63.85 | 43.58 | +20.27 | 31.59 | 28.96 | +2.63 |

Table 4 displays how most cases of non-improvement occur in DTD datasets. Although the differences are not large, they occur with different detectors. It is also essential to mention that in this same dataset, where the other detectors did not perform so well (e.g., Mahalanobis, SSD and KNN), they have improved considerably with our approach. So for some cases, our approach con-

Table 4: AUROC values for different approaches (i.e., no augmentation vs 32). Results obtained with ResNet18 model trained with CIFAR-10 dataset.

| detectors | SVHN | | | iSUN | | | CIFAR-100 | | | LSUN | | | DTD | | | Places365 | | |
|---|---|---|---|---|---|---|---|---|---|---|---|---|---|---|---|---|---|---|
| | no | 32 | Impr. | no | 32 | Impr. | no | 32 | Impr. | no | 32 | Impr. | no | 32 | Impr. | no | 32 | Impr. |
| energy | 97.32 | 97.63 | +0.31 | 94.47 | 95.12 | +0.65 | 88.18 | 89.10 | +0.92 | 96.91 | 97.27 | +0.36 | 97.29 | 96.66 | −0.63 | 96.16 | 96.43 | +0.27 |
| nnguide | 96.74 | 97.19 | +0.45 | 92.24 | 93.15 | +0.91 | 84.48 | 85.10 | +0.62 | 95.65 | 96.00 | +0.35 | 97.58 | 97.10 | −0.48 | 94.79 | 95.01 | +0.22 |
| msp | 95.11 | 95.94 | +0.83 | 92.29 | 93.17 | +0.88 | 88.35 | 89.35 | +1.00 | 94.64 | 95.41 | +0.77 | 91.11 | 90.77 | −0.34 | 93.94 | 94.58 | +0.64 |
| maxlogit | 97.17 | 97.54 | +0.37 | 94.34 | 95.01 | +0.67 | 88.19 | 89.14 | +0.95 | 96.76 | 97.16 | +0.40 | 97.05 | 96.41 | −0.64 | 96.02 | 96.32 | +0.30 |
| ssd | 96.85 | 97.30 | +0.45 | 69.03 | 69.05 | +0.02 | 76.18 | 77.18 | +1.00 | 88.62 | 89.21 | +0.59 | 88.79 | 97.78 | +8.99 | 88.87 | 89.58 | +0.71 |
| mahalanobis | 97.39 | 97.78 | +0.39 | 77.92 | 78.33 | +0.41 | 80.50 | 81.50 | +1.00 | 91.66 | 92.24 | +0.58 | 88.76 | 97.85 | +9.09 | 91.71 | 92.31 | +0.60 |
| knn | 96.05 | 96.56 | +0.51 | 94.26 | 94.87 | +0.61 | 89.95 | 90.67 | +0.72 | 95.57 | 96.10 | +0.53 | 73.17 | 85.00 | +11.83 | 95.03 | 95.39 | +0.36 |

verges towards the detection performance of traditional inference, but, in the majority of the cases, it improves results considerably.

**Accuracy evaluation**. The employed model results in an accuracy of 94.24%. This value is obtained with the validation dataset of Cifar-10 with no augmentation technique applied (i.e., as commonly calculated). However, as in our approach we employ augmentation techniques during inference, the accuracy is also affected. Replica 1 resulted in an accuracy of 94.12, 8 replicas resulted in an accuracy of 94.17, and 32 replicas in an accuracy of 94.20. Hence, with these results, we can not state that increasing the number of replicas or using augmented replicas have a statistically significant impact on the accuracy.

## 3.2 ID: PATHMNIST AND MOBILENETV2

**Model architecture**. For the second analysis, we use MobileNetV2 architecture for the model. This different model architecture ensures that the contribution is not only viable for ResNets. Furthermore, the model is trained with PathMNIST dataset (Yang et al., 2023) from the MedMNIST collection. MedMNIST comprises a series of datasets from the medical field. PathMNIST dataset offers two advantages in our analysis. Firstly, the source images are sized $3 \times 224 \times 224$, allowing us to examine if the proposed approach is viable for larger images. Secondly, the medical domain is particularly interested in being able to classify images with certainty. Besides, the timing cost that entails our approach, as it requires inferring a larger set of images, is not critical for a high number of use cases (e.g., diagnostics). The MobileNetV2 model has been trained through 100 epochs, with a resulting accuracy of 94.33%.

**OOD datasets**. In addition, the model is evaluated with different OOD datasets: Describable Textures Dataset (DTD) (Cimpoi et al., 2014), Places365 (Zhou et al., 2017), SUN (Xiao et al., 2010), iNaturalist (Van Horn et al., 2018).

**Augmentation technique**. The images are resized to a fixed size, which is required by OOD data sets with different sizes. Center crop of size $192 \times 192$ pixels is performed for the resize. The image is randomly flipped horizontally with a probability of 0.5. Random Rotation until 30 degrees is also applied. Finally, AugMix technique applies a combination of augmentations with a severity level of 5 to enhance the diversity of the replicas. Note that Hendrycks et al. (2019) proposed AugMix with the aim of *"improving robustness and uncertainty"*. The transformation techniques used in inference are the same ones used in the training process.

It is noteworthy to mention that different transformation techniques (compared to Section 3.1) are used to ensure that our approach is not only valid for specific transformations. Besides,

**Obtained results employing augmented replicas**. In this analysis, we obtain similar results and conclusions to the previous experiment. Tables 5 and 6 provide the results of AUROC and FPR95 for the different replicas (i.e., 1, 8 and 32) and datasets.

Results from Tables 5 and 6 show that the performance of the detectors is improved by using transformed replicates. On the one hand, FPR95 decreases in all presented scenarios, with no exception. On the other hand, AUROC value increases when we employ a higher number of replicas. Thereby, all employed detectors improve in all datasets with our approach.

Figure 3 shows the mean value of the detectors using different number of replicas. In this case, we evaluate it with 1, 8 and 32 replicas. The image clearly shows that both AUROC and FRP95 are

Table 5: FPR95 values for different detectors, datasets and number of replicas (i.e., 1, 8 and 32). Results obtained with MobileNetV2 model trained with PathMNIST dataset.

| detectors | iNaturalist | | | SUN50 | | | DTD | | | Places50 | | |
|---|---|---|---|---|---|---|---|---|---|---|---|---|
| | 1 | 8 | 32 | 1 | 8 | 32 | 1 | 8 | 32 | 1 | 8 | 32 |
| energy | 24.41 | 9.62 | 9.09 | 29.54 | 14.91 | 14.52 | 25.62 | 15.78 | 15.37 | 32.81 | 17.43 | 17.28 |
| nnguide | 15.28 | 4.90 | 4.28 | 22.26 | 10.39 | 9.76 | 18.14 | 11.81 | 11.13 | 25.83 | 12.37 | 11.30 |
| msp | 65.30 | 48.97 | 44.43 | 65.95 | 47.56 | 43.07 | 70.48 | 54.75 | 51.12 | 68.64 | 49.77 | 45.14 |
| maxlogit | 28.59 | 11.77 | 11.07 | 32.73 | 16.60 | 16.04 | 29.34 | 17.22 | 16.81 | 36.39 | 19.29 | 18.53 |
| ssd | 29.59 | 7.06 | 6.62 | 35.95 | 6.94 | 6.38 | 23.58 | 7.98 | 7.93 | 42.71 | 9.75 | 8.82 |
| mahalanobis | 28.32 | 6.32 | 5.88 | 33.60 | 6.32 | 5.52 | 21.67 | 7.36 | 7.11 | 40.77 | 9.10 | 7.90 |
| knn | 58.09 | 43.45 | 41.87 | 58.53 | 40.97 | 39.83 | 54.38 | 44.61 | 43.58 | 63.43 | 46.08 | 43.95 |

Table 6: AUROC values for different detectors, datasets and number of replicas (i.e., 1, 8 and 32). Results obtained with MobileNetV2 model trained with PathMNIST dataset.

| detectors | iNaturalist | | | SUN50 | | | DTD | | | Places50 | | |
|---|---|---|---|---|---|---|---|---|---|---|---|---|
| | 1 | 8 | 32 | 1 | 8 | 32 | 1 | 8 | 32 | 1 | 8 | 32 |
| energy | 96.11 | 98.00 | 98.12 | 95.09 | 97.25 | 97.38 | 94.89 | 96.52 | 96.66 | 94.61 | 96.96 | 97.12 |
| nnguide | 96.84 | 98.41 | 98.51 | 95.89 | 97.80 | 97.90 | 95.77 | 96.99 | 97.10 | 95.37 | 97.49 | 97.61 |
| msp | 89.38 | 92.88 | 93.51 | 89.35 | 92.97 | 93.55 | 87.22 | 90.20 | 90.77 | 88.59 | 92.41 | 93.06 |
| maxlogit | 95.68 | 97.77 | 97.90 | 94.76 | 97.09 | 97.23 | 94.42 | 96.25 | 96.41 | 94.26 | 96.77 | 96.95 |
| ssd | 95.42 | 97.95 | 98.07 | 94.92 | 97.86 | 98.00 | 95.81 | 97.73 | 97.78 | 94.17 | 97.52 | 97.70 |
| mahalanobis | 95.57 | 98.01 | 98.12 | 95.11 | 97.93 | 98.07 | 95.98 | 97.80 | 97.85 | 94.40 | 97.60 | 97.78 |
| knn | 86.70 | 90.07 | 90.41 | 87.83 | 91.47 | 91.77 | 82.16 | 84.69 | 85.00 | 86.54 | 90.37 | 90.82 |

improving. Furthermore, they show how that improvement reaches an asymptote. The improvement is significant from 1 to 8 but slows down from 8 to 32 replicates. Therefore, we conclude that there is little to gain from more than 32 replicas for this case (i.e., model, training dataset, transformation techniques).

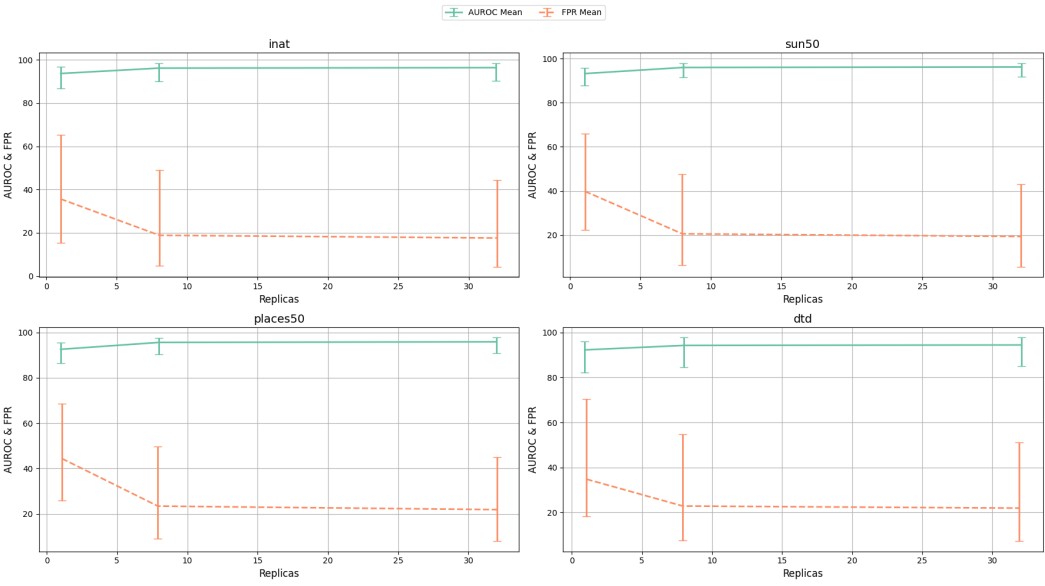

Figure 3: Mean FPR95 and mean AUROC of all detectors through different OOD datasets. Results obtained with MobileNetV2 model trained for PathMNIST dataset.

**Non Augmented vs Augmented inference**. Tables 3 and 4 displays the obtained results. Similar to Cifar-10 results, Tables 3 and 4 report that there are only a few cases where there is no improvement. Furthermore, there are no cases with a significant deteriorate.

Table 7: FPR95 values for different approaches (i.e., no augmentation vs 32). Results obtained with MobileNetV2 model trained with PathMNIST dataset.

| detectors | iNaturalist | | | SUN50 | | | DTD | | | Places50 | | |
|---|---|---|---|---|---|---|---|---|---|---|---|---|
| | no | 32 | Impr. | no | 32 | Impr. | no | 32 | Impr. | no | 32 | Impr. |
| energy | 6.07 | 9.09 | −3.02 | 15.74 | 14.52 | +1.22 | 12.36 | 15.37 | −3.01 | 18.14 | 17.28 | +0.86 |
| nnguide | 4.60 | 4.28 | +0.32 | 11.88 | 9.76 | +2.12 | 8.88 | 11.13 | −2.25 | 13.68 | 11.30 | +2.38 |
| msp | 47.28 | 44.43 | +2.85 | 56.69 | 43.07 | +13.62 | 57.82 | 51.12 | +6.70 | 57.87 | 45.14 | +12.73 |
| maxlogit | 7.79 | 11.07 | −3.28 | 18.14 | 16.04 | +2.10 | 14.41 | 16.81 | −2.40 | 20.55 | 18.53 | +2.02 |
| ssd | 64.49 | 6.62 | +57.87 | 81.69 | 6.38 | +75.31 | 45.32 | 7.93 | +37.39 | 82.37 | 8.82 | +73.55 |
| mahalanobis | 64.36 | 5.88 | +58.48 | 81.05 | 5.52 | +75.53 | 44.96 | 7.11 | +37.85 | 81.73 | 7.90 | +73.83 |
| knn | 74.55 | 41.87 | +32.68 | 85.17 | 39.83 | +45.34 | 63.85 | 43.58 | +20.27 | 85.97 | 43.95 | +42.02 |

Table 8: AUROC values for different approaches (i.e., no augmentation vs 32). Results obtained with MobileNetV2 model trained with PathMNIST dataset.

| detectors | iNaturalist | | | SUN50 | | | DTD | | | Places50 | | |
|---|---|---|---|---|---|---|---|---|---|---|---|---|
| | no | 32 | Impr. | no | 32 | Impr. | no | 32 | Impr. | no | 32 | Impr. |
| energy | 98.46 | 98.12 | −0.34 | 97.25 | 97.38 | +0.13 | 97.29 | 96.66 | −0.63 | 96.92 | 97.12 | +0.20 |
| nnguide | 98.44 | 98.51 | +0.07 | 97.46 | 97.90 | +0.44 | 97.58 | 97.10 | −0.48 | 97.05 | 97.61 | +0.56 |
| msp | 93.47 | 93.51 | +0.04 | 92.09 | 93.55 | +1.46 | 91.11 | 90.77 | −0.34 | 91.56 | 93.06 | +1.50 |
| maxlogit | 98.27 | 97.90 | −0.37 | 97.03 | 97.23 | +0.20 | 97.05 | 96.41 | −0.64 | 96.68 | 96.95 | +0.27 |
| ssd | 85.83 | 98.07 | +12.24 | 78.81 | 98.00 | +19.19 | 88.79 | 97.78 | +8.99 | 78.47 | 97.70 | +19.23 |
| mahalanobis | 85.71 | 98.12 | +12.41 | 78.86 | 98.07 | +19.21 | 88.76 | 97.85 | +9.09 | 78.54 | 97.78 | +19.24 |
| knn | 74.35 | 90.41 | +16.06 | 69.23 | 91.77 | +22.54 | 73.17 | 85.00 | +11.83 | 69.18 | 90.82 | +21.64 |

**Accuracy evaluation**. The accuracy obtained evaluating the validation dataset with no augmentation is 94.33. If we employ transformed images and replicas, the accuracies obtained are 92.42 (one replica), 92.12 (8 replicas) and 92.25 (32 replicas).

## 4 RELATED WORK

In the realm of OOD detectors for Neural Networks (NNs), several noteworthy methods have been proposed. MSP method by Hendrycks & Gimpel (2016), leverages probabilities from softmax distributions, observing that correctly classified examples often have greater maximum softmax probabilities than erroneously classified and OOD examples. The MaxLogit method by Wei et al. (2022), addresses the overconfidence issue in neural networks by enforcing a constant vector norm on the logits during training. The Mahalanobis method utilizes the Mahalanobis distance for OOD detection (Lee et al., 2018). The Self-Supervised Outlier Detection (SSD) method employs self-supervised representation learning (Sehwag et al., 2021) followed by a Mahalanobis distance-based detection in the feature space. The GradNorm method uses information extracted from the gradient space for OOD detection (Huang et al., 2021). The KNN method employs non-parametric nearest-neighbor distance for OOD detection (Sun et al., 2022). The Energy-based method uses an energy score for OOD detection, distinguishing in- and out-of-distribution samples more effectively than traditional softmax scores (Liu et al., 2020). Lastly, the Nearest Neighbor Guidance (NNGuide) method guides the classifier-based score to respect the boundary geometry of the data manifold, reducing the overconfidence of OOD samples while preserving the fine-grained capability of the classifier-based score (Park et al., 2023). These methods represent the current state-of-the-art in OOD detection for NN.

## 5 CONCLUSIONS

This paper proposes to use augmented replicas during inference to improve OOD detectors. This work offers a simple but effective contribution. By simply replicating images and transforming them before inference, we are able to improve OOD detection in the vast majority of the cases. In an abstract, one can say that we allow the OOD detectors to view the inputs from slightly different perspectives. This approach enhances the "knowledge" they may gain, including determining whether an input is OOD. Furthermore, the contribution is based on simple statistics such as the median,

but it is also open to more elaborate statistical methods. Therefore, with the presented results, we conclude that our proposal is a contribution to the field of trustworthy AI.

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

## A    APPENDIX:

In the data and results presented throughout this article, the augmentation used for inference was the same as that used during the training process. However, in this section, we also evaluate the results of the two models with varying levels of image transformations during inference. Since the models were not retrained, the transformation level during training remains unchanged, e.g., 5 pixel flips in CIFAR-10 and AugMix severity level of 5. For inference, some of the transformation parameters are modified.

**Random Pixel Flip** For the CIFAR-10 model, trained with 5 Random Pixel Flips, we evaluated its performance with different numbers of pixel flips. Figure 4 shows the FPR95 and AUROC data for two detectors as the image noise increases, i.e., with an increasing number of Random Pixel Flips. The Out-of-Distribution (OOD) data used in this experiment is taken from the CIFAR-100 dataset (near OOD).

The graph indicates that as the noise in the images increases, both FPR95 and AUROC improve. However, there is a point where the noise becomes counterproductive, making it difficult to identify the discriminating characteristics of the images. Depending on the measurement and the detector, the optimal number of Random Pixel Flips varies, but it generally falls within the range of 8 to 32.

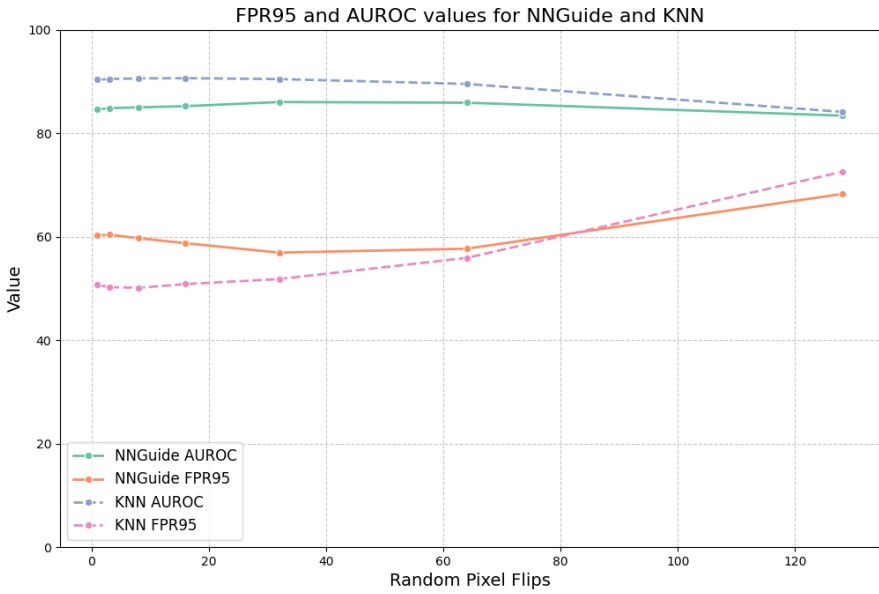

Figure 4: FPR95 and AUROC for NNGuide and KNN with different numbers of Pixel Flips transformation. Results predictions are obtained with the median value of 8 replicas and Cifar-100 OOD.

**AugMix** Figure 5 depicts the results with different AugMix severity levels. The graph presents the results for SUN50 OOD. In this case, NNGuide results do not vary significantly with different severities. However, for KNN, increasing severity is counterproductive.

In summary, these data show that identifying the optimal technique is not straightforward. The variation in results depends on several factors, including the model, detector, ID data, OOD data, transformation, and the level of transformation.

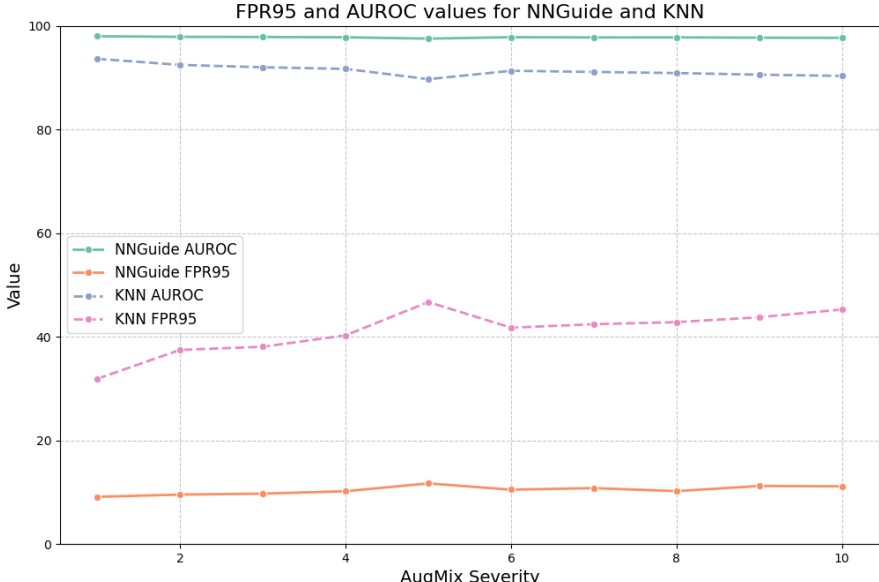

Figure 5: FPR95 and AUROC for NNGuide and KNN with different severity of AugMix transformation. Results predictions are obtained with the median value of 8 replicas and SUN50 OOD.

These results suggest that the approach allows for manipulation of the transformation level. However, further study is needed to determine which transformations are effective and how to select the optimal levels.

