# OpenReview forum: "Power of Augmented Replicas in Out-Of-Distribution Detection"
_ICLR.cc/2025/Conference — Submitted to ICLR 2025_

### Official Review · Reviewer_QVmm · 2024-10-25

**Soundness:** 2
**Presentation:** 2
**Contribution:** 1
**Rating:** 3
**Confidence:** 5

**Summary:**

This paper uses image augmentation strategy during inference to improve OOD detection. Image augmentation brings performance improvement.

**Strengths:**

This paper does not require training and can improve the performance of OOD Detection using a very simple method.

**Weaknesses:**

This paper is not promising enough for contributing to the related domain and community. Here I will list the weaknesses of the paper:

* The core design motivation of this paper is insufficient. This article does not analyze the advantages and disadvantages of current OOD detection methods well, and does not clearly explain why the designed method is proposed.

* Based on the weak motivation, the method in this paper is more like an engineering trick than a research contribution. The author believes that the method brings more "knowledge" about the data to the model, but there is no explanation, reasoning, or experimental results to show what this knowledge is.

* Why did the performance increase? This paper does not explain this question. There is not even any qualitative and quantitative experimental support for the specific design of data augmentation details and related ablation experiments.

* The writing and presentation of this paper is not very clear and there are some errors. I hope the author can check it carefully. (For example, the last element of each matrix in Figure 1 should be f_xn instead of f_4n)

**Questions:**

I suggest that the author could consider the following questions:

* What is the essence of OOD Detection? Is it the expansion of knowledge? What knowledge does a trained model (space) need?

* If training-free is a good premise, what is the relationship between generalization across different distributions and data augmentation?

* What does the data augmentation proposed in the article actually do for feature distribution?

---

### Official Review · Reviewer_QBaJ · 2024-11-02

**Soundness:** 2
**Presentation:** 3
**Contribution:** 1
**Rating:** 3
**Confidence:** 4

**Summary:**

I recommend rejecting this paper due to several concerns. First, it employs well-known and widely used techniques (e.g., test-time augmentation) on an existing task (OOD detection), without introducing significant novelty. Additionally, a previously published study presents directly opposing results, which the authors have not addressed or discussed—an essential omission in establishing the validity of their findings. Furthermore, the experimental setup in this paper raises questions about reliability and reproducibility. For these reasons, I believe a rejection is warranted.

**Strengths:**

The paper presents the proposed method with good clarity, making it very easy to follow and understand.

**Weaknesses:**

The paper lacks an in-depth understanding of why test-time augmentation is effective in this context. Please refer to the "Question" section for further details.

Additionally, the novelty of the work is limited as it use test time data augmentation directly without a specific design on OOD detection.

The statistical significance analysis in Figure 2 appears questionable, especially on the iSUN dataset. It seems as though the model may not be learning effectively.

**Questions:**

My primary concern with this paper is the lack of clarity on why test-time augmentation works effectively for the OOD detection task. This should be the central focus, yet it remains insufficiently addressed. The findings presented here appear inconsistent with those in [1], where test datasets created with data augmentation techniques showed a significant performance drop as more and stronger augmentations were applied. This aligns with existing literature suggesting that data augmentation often increases the domain gap from the original data, thereby leading to performance degradation in OOD tasks [2,3].

The authors should carefully reassess their claims and clearly specify which augmentation techniques are beneficial for OOD detection in this context.

References:

[1] Unsupervised Evaluation of Out-of-distribution Detection: A Data-centric Perspective, arXiv 2023.

[2] Affinity and Diversity: Quantifying Mechanisms of Data Augmentation, ICLR 2022.

[3] Are Labels Always Necessary for Classifier Accuracy Evaluation? CVPR 2021.

**Details Of Ethics Concerns:**

This paper use existing datasets. Thus no ethics concerns.

---

> ### Comment · Reviewer_QBaJ · 2024-11-27
> **No rebuttal**
>
> It appears no rebuttal was provided by the authors, thus I will keep my current rating.

---

### Official Review · Reviewer_4d8L · 2024-11-04

**Soundness:** 2
**Presentation:** 1
**Contribution:** 1
**Rating:** 3
**Confidence:** 5

**Summary:**

This paper explores the potential of utilizing data augmentations to enhance existing out-of-distribution (OOD) detection techniques. The study shows that using the median score across different augmentations improves OOD detection performance.

**Strengths:**

The paper demonstrate that on CIFAR-10 and PathMNIST datasets, utilizing data augmentations and the median score reliably improves on existing OOD detection metrics.

**Weaknesses:**

1. The paper has major presentation issues:
- Figure 2 plots AUROC and FPR95 on the same plot. This makes the information very difficult to digest because these two metrics are anti-correlated. Additionally, the range of each values are on opposite ends of the spectrum (FPR95 closer to 0 while AUROC closer to 100), making the performance change caused by increase in number of replicas illegible.
- It is unclear what information Figure 2 is suppose to convey in addition to Table 1 and 2. It also isn't very information-rich since Figure 2 x-axis has large range from 0 to over 30, yet only 3 data points is plotted.
- Table 1 and 2 is a huge table of numbers without any bolding. It is unclear what the key takeaway is.

2. The datasets and model used for evaluation is not comprehrensive:
- OOD detection is either only evaluated on CIFAR-10, which is low resolution, or PathMNIST, which is a pathology dataset. A more common large-scale image recognition dataset used for evaluation is ImageNet.
- Results are only reported on MobileNetV2 architecture, which is an atypical choice, making it difficult to compare with existing literature. More common architecture includes: ResNet-50, ResNet-101, Wide-ResNet-101, ViT-L/14.

3. Lack of comment on existing literature that augments model predictions using data augmentations. [1] demonstrates that data augmentation can be very helpful for the task of novelty prediction, which is very similar to OOD detection.

[1] Bahat, Yuval, and Gregory Shakhnarovich. "Confidence from invariance to image transformations." arXiv preprint arXiv:1804.00657 (2018).

**Questions:**

1. Any particular reason why the median is used rather than the typical estimate of mean? Is the resulting score distribution too skewed?
2. Is there any patterns that merges regarding the augmentation used for the median score? i.e. what kind of augmentation typically lead to the median score? It would be interesting if it is a particular transformation or equally interesting if it is random and no pattern emerges.
3. Curious if there are any thoughts on whether or not the increase compute (extra passes through the model) is worth the increased detection performance.

---

### Official Review · Reviewer_v52C · 2024-11-04

**Soundness:** 2
**Presentation:** 2
**Contribution:** 2
**Rating:** 3
**Confidence:** 3

**Summary:**

This paper proposes a novel approach to improve  OOD detection by employing multiple augmented replicas of a single inference image. specifically, by replicating and transforming inference images, then calculating a median score across these replicas to assess the OOD prediction.

**Strengths:**

The authors validate the approach across multiple detectors, models, and datasets, including different types of transformations, to show that the method is effective, the approach is both straightforward and computationally feasible.

**Weaknesses:**

1. This approach feels more like an engineering trick than a genuinely novel method, as similar techniques are commonly used in tasks like image segmentation and classification.
2.  The increase in computational demand does not appear proportionate to the improvements in detection accuracy.
3. The method has not been validated on larger datasets, such as ImageNet.

**Questions:**

See Weakness

---

### Meta-Review · Area_Chair_Rfa7 · 2024-12-20

**Metareview:**

This paper proposes a method for OOD detection.  The strategy is to use the median score across many augmented versions of the input sample instead of a single score on the original sample.

Strengths: the approach is simple.  The experiment validate the approach on different detectors, models and datasets.

Weaknesses: Reviewers raised several concerns ranging from poor motivation and justification, poor presentation and insufficient experimentation on larger-scale datasets.  A key weakness, given the simplicity of the methodology, would be exploration on the reason behind the effectiveness.

After reading the reviews, the AC has decided to reject the paper.  This decision is in line with all four reviewers who recommend that the paper be rejected.

**Additional Comments On Reviewer Discussion:**

The authors did not provide any response and there was no further discussion.

---

### Decision · Program_Chairs · 2025-01-22

Reject